

# Aboveground and belowground arthropods experience different relative influences of stochastic versus deterministic community assembly processes following disturbance

Scott Ferrenberg[1], Alexander S. Martinez[1,2] and Akasha M. Faist[1]

[1] Department of Ecology and Evolutionary Biology, University of Colorado at Boulder, Boulder, CO, United States
[2] Department of Biological Sciences, Purdue University, West Lafayette, IN, United States

## ABSTRACT

**Background**. Understanding patterns of biodiversity is a longstanding challenge in ecology. Similar to other biotic groups, arthropod community structure can be shaped by deterministic and stochastic processes, with limited understanding of what moderates the relative influence of these processes. Disturbances have been noted to alter the relative influence of deterministic and stochastic processes on community assembly in various study systems, implicating ecological disturbances as a potential moderator of these forces.

**Methods**. Using a disturbance gradient along a 5-year chronosequence of insect-induced tree mortality in a subalpine forest of the southern Rocky Mountains, Colorado, USA, we examined changes in community structure and relative influences of deterministic and stochastic processes in the assembly of aboveground (surface and litter-active species) and belowground (species active in organic and mineral soil layers) arthropod communities. Arthropods were sampled for all years of the chronosequence via pitfall traps (aboveground community) and modified Winkler funnels (below-ground community) and sorted to morphospecies. Community structure of both communities were assessed via comparisons of morphospecies abundance, diversity, and composition. Assembly processes were inferred from a mixture of linear models and matrix correlations testing for community associations with environmental properties, and from null-deviation models comparing observed vs. expected levels of species turnover (Beta diversity) among samples.

**Results**. Tree mortality altered community structure in both aboveground and belowground arthropod communities, but null models suggested that aboveground communities experienced greater relative influences of deterministic processes, while the relative influence of stochastic processes increased for belowground communities. Additionally, Mantel tests and linear regression models revealed significant associations between the aboveground arthropod communities and vegetation and soil properties, but no significant association among belowground arthropod communities and environmental factors.

**Discussion**. Our results suggest context-dependent influences of stochastic and deterministic community assembly processes across different fractions of a spatially

Corresponding author
Scott Ferrenberg,
sferrenberg@usgs.gov,
scott.ferrenberg@colorado.edu

co-occurring ground-dwelling arthropod community following disturbance. This variation in assembly may be linked to contrasting ecological strategies and dispersal rates within above- and below-ground communities. Our findings add to a growing body of evidence indicating concurrent influences of stochastic and deterministic processes in community assembly, and highlight the need to consider potential variation across different fractions of biotic communities when testing community ecology theory and considering conservation strategies.

## INTRODUCTION

Understanding the processes governing the assembly of biotic communities is a longstanding goal in ecology. Deterministic processes have long been considered primary drivers of biodiversity patterns and niche-based theories of community assembly have amassed substantial support (e.g., *MacArthur, 1957*; *Tilman, 1982*). In contrast, theories proposing that stochastic processes can shape community structure—largely independent of species' traits—have also received support (*MacArthur & Wilson, 1967*; *Connell, 1978*; *Hubbell, 2001*; *Chave, 2004*; *Adler, HilleRisLambers & Levine, 2007*). Despite the apparent contradiction in theories, recent work has revealed simultaneous influences of deterministic and stochastic processes in the assembly and structure of a diverse range of biotic communities (*Hart, 1992*; *Thompson & Townsend, 2006*; *Cadotte, 2007*; *Chase, 2007*; *Ellwood, Manica & Foster, 2009*; *Rominger, Miller & Collins, 2009*; *Lepori & Malmqvist, 2009*; *Fišer, Blejec & Trontelj, 2012*). As evidence of a concurrent influence of deterministic and stochastic assembly processes mounts, it also raises a key question: *what determines the relative influence of stochastic and deterministic processes in community assembly*?

Experimental evidence and theory have implicated a suite of factors controlling the relative influence of deterministic and stochastic processes in biotic communities—e.g., ecosystem productivity, regional biodiversity and dispersal rates, habitat connectivity, species' interactions and priority effects, and ecosystem disturbances (*Chase, 2003*; *Chase, 2007*; *Jiang & Patel, 2008*; *Collinge & Ray, 2009*; *Lepori & Malmqvist, 2009*; *Vergnon, Dulvy & Freckleton, 2009*; *Stokes & Archer, 2010*). Of these factors, disturbances have been reported to increase (*Chase, 2007*; *Jiang & Patel, 2008*) and decrease (*Didham, Watts & Norton, 2005*; *Leibold & McPeek, 2006*) the relative influence of both deterministic and stochastic processes, with recent work indicating that the importance of deterministic and stochastic processes can shift over time following disturbance (*Lepori & Malmqvist, 2009*; *Ferrenberg et al., 2013*; *Nemergut et al., 2013*). Evidence also indicates that assembly processes can vary among different fractions of a community in relation to environmental gradients, as well as species' ecological strategies, relative abundances, and dispersal rates (*Thompson & Townsend, 2006*; *Kraft, Valencia & Ackerly, 2008*; *Ellwood, Manica & Foster, 2009*; *Rominger, Miller & Collins, 2009*; *Barber & Marquis, 2011*; *Langenheder & Székely,*

*2011*; *Ingwell et al., 2012*; *Armitage, Ho & Quigg, 2013*; *Márquez & Kolasa, 2013*; *Arnan, Cerdá & Retana, 2015*; *Guo et al., 2014*). Understanding how ecological disturbances interact with these mechanisms to influence the strength of stochastic versus deterministic processes across different fractions of communities is an important next step for community assembly theory.

Ground-dwelling arthropod communities are ideal for the study of community assembly processes as they are composed of taxa representing a diverse range of ecological strategies and dispersal capabilities (*Speight, Hunter & Watt, 2008*). Ground-dwelling arthropods in forested systems are also generally sensitive to a range of disturbance types and intensities, offering the chance to explore the effects of disturbance on assembly processes across different fractions of these communities (*Ferrenberg et al., 2006*; *Moretti, Duelli & Obrist, 2006*; *Lessard et al., 2011*; *Ober & DeGroote, 2011*; *Arnan et al., 2013*; *Delph et al., 2014*; *Williams et al., 2014*; *Brunbjerg et al., 2015*). We used the opportunity presented by a multi-year bark beetle infestation to investigate the effects of tree mortality on assembly processes and community structure in ground-dwelling arthropod communities. We captured temporal variation by substituting space for time along a five-year chronosequence of tree mortality from bark beetles in a subalpine forest of the southern Rocky Mountains. Previous work indicates that bark beetle-induced tree mortality can rapidly alter understory and soil environments through changes in microclimate (*Wiedinmyer et al., 2012*; *Maness, Kushner & Fung, 2013*), soil hydrology (*Mikkelson et al., 2013*), soil nutrient pools (*Morehouse et al., 2008*; *Griffin, Turner & Simard, 2011*; *Xiong et al., 2011*; *Griffin & Turner, 2012*), and understory plant productivity (*Brown et al., 2010*). Thus, we hypothesized (1) that tree mortality would alter arthropod community structure over time, and (2) that changes in arthropod community structure would be linked to deterministic influences, likely from influences of changing understory vegetation cover and soil environments. Finally, substantial variation in the ecological strategies and dispersal potential exists between aboveground arthropods (active on the ground surface and in upper litter layers) and belowground arthropods (active in organic and mineral soil layers) (*Blossey & Hunt-Joshi, 2003*; *De Deyn & Van der Putten, 2005*; *Joern & Laws, 2013*). Thus, we hypothesized (3) that aboveground arthropods, which we assumed to have greater mobility and thus greater ability to track changing environments, would exhibit stronger associations to local environmental properties, while belowground arthropods would exhibit weaker associations to the environment due to dispersal limitations.

## METHODS

### Study site and chronosequence

We characterized arthropod communities, vegetation cover, and soil properties across a five year chronosequence of tree mortality previously described in a study of soil bacteria by *Ferrenberg et al. (2014)*. Year zero (chronosequence year 0) represented samples from under living trees that were never attacked by bark beetles, with remaining samples coming from four categories representing trees killed by bark beetles one to four years prior to our study (chronosequence years 1–4). All sampled plots of the chronosequence were

located under mature limber pines (*Pinus flexilis*) at the University of Colorado's Mountain Research Station, 2,900 m above sea level and approximately 11 km east of the Continental Divide in Boulder County, Colorado, USA (40°N; 105°W). This site is characterized by low average annual temperatures and a majority of annual precipitation falls as snow during winter months (*Mitton & Ferrenberg, 2012*; *Duhl et al., 2013*; *Ferrenberg et al., 2014*). Tree mortality caused by the mountain pine beetle (*Dendroctonus pondersae*) began in this site in 2006 and continued through 2012 in susceptible pines that were monitored monthly allowing the establishment of the chronosequence used here (*Ferrenberg et al., 2014*; *Ferrenberg, Kane & Mitton, 2014*; *Ferrenberg & Mitton, 2014*). This site is now characterized by a mosaic of living trees and trees in variable states of decay.

## Arthropod, vegetation, and soil sampling

Arthropods were sampled twice (June and August) to characterize communities during the three month period when understory vegetation of this sub-alpine ecosystem is productive. We sampled surface-dwelling arthropods (aboveground arthropods) from under 40 focal trees using a combination of two pitfall traps per tree (i.e., 80 pitfall traps in total), with each trap placed approximately one meter from the focal tree's trunk. Focal trees were surrounded by trees with similar health status, were bounded from other sampling locations by live trees, and were evenly divided among the five years of the insect-induced tree mortality chronosequence (i.e., eight sample plots per each of the five chronosequence years, with the exception of the chronosequence year four where both pitfall traps were removed from one plot by an animal during the experiment leaving seven replicates). Focal trees were selected from a larger number of trees within each chronosequence year to avoid geographical clumping and maximize the distance between sampling areas. Pitfall traps from under each focal tree were separated from other traps by a minimum of 15 m, while traps from within the same chronosequence year were separated by ≥20 m; distances as little as 5 m between pitfall traps have been verified as being methodologically valid approaches for sampling arthropods in mixed woodland ecosystems via *Ward, New & Yen (2001)*. Pitfall traps were 225 ml plastic sample cups (8 cm deep × 6 cm diameter) that were inserted into organic and mineral soils with their lip flush to the ground surface. Each trap contained 80 ml of soapy water to act as a killing agent and preservative. Pitfalls were left open for 72 h in mid-June and another 72 h in early-August 2011. At the end of each sampling period, the traps were drained of excess soapy water, filled with 80% EtOH and stored at −4 °C until arthropods were sorted to morphospecies and counted. Arthropods primarily found belowground in soil and organic layers were sampled from under 30 focal trees, six trees per each of the five chronosequence years, via modified Winkler extractors. Samples for Winkler extractors were collected in June and August by cutting a 10 cm diameter soil/litter plug to a depth of 8 cm in the mineral soil and extracting an undisturbed column. Three column samples, evenly spaced under each focal tree (one meter from the trunk) were composited together in plastic bags in the field, returned to the lab within two hours, and placed into Winkler extractors held under 80 watt lamps for 5 days. Collection cups for each extractor contained a 1:1 solution of EtOH (100%) and distilled $H_2O$ as a killing agent and preservative. The cups were capped and

stored at $-4$ °C until samples were sorted to morphospecies and counted. The majority of arthropods captured were either adult holometabolous hexapods (i.e., beetles and ants), or adult to late-stage instars of hemimetabolous hexapods and arachnids (i.e., collembola, mites, and spiders). All captured individuals were sorted to morphospecies and identified to order, with hexapods further identified to families. The effectiveness of trapping effort at characterizing the aboveground and belowground community was assessed via species accumulation curves created in PC-ORD. In addition to arthropod sampling, cover by plant functional groups (herbaceous plants, grasses, woody plants), vegetation species richness, and surface rock cover were measured at peak biomass in a circular plot (1 m radius from each tree's trunk, or an area of roughly 4.1 m$^2$) placed around the trunk of each focal tree. All trees used in the study were of similar size, but data for each tree was nevertheless corrected for small variations in tree size by converting all aerial cover estimates to value per m$^2$ of ground surface surveyed.

Measures of soil chemical properties from under each focal tree were completed in the spring of 2011, prior to any plot disturbances due to arthropod sampling. Soil samples were a composite of three, 130.5 cm$^3$ cores from the top 5 cm of mineral soil (with all litter and visible organic materials removed) collected evenly from around the tree and roughly 1.25 m from the trunk. Following field extraction, all samples were transported on ice, and sieved through 2 mm mesh before biogeochemical analyses. Soil moisture, pH, total %C and %N, C:N ratio, $NH_4^+$, dissolved organic carbon (DOC), and microbial biomass were quantified using the detailed methods described in *Ferrenberg et al. (2013)* and *Ferrenberg et al. (2014)*. In brief, soil moisture was determined via gravimetric dry-down, pH was measured from a 1:5 ratio of soil to distilled and de-ionized $H_2O$, and total C and N were determined using combustion. Measures of $NH_4^+$, DOC, and microbial biomass were determined via extractions from soil with 0.5 M $K_2SO_4$. Concentration of $NH_4^+$ was determined from absorbance on a microplate reader, while DOC was determined using a TIC/TOC analyzer, with DOC = EC/kEC where EC = extractable C from soil and kEC = extractable C from microbial biomass (*Beck et al., 1997*). Soil chemistry data from the field site are available from figshare (*Knelman, 2014*).

## Data analysis

June and August arthropod samples were binned into one grand sample per focal tree prior to analyses to match the primary goal of investigating assembly processes among years following disturbance, as opposed to across a growing season. We then used non-metric multidimensional scaling (NMDS) to visualize the community structure (at the level of morphospecies) of above and belowground arthropods, and one-way PERMANOVA (followed by pairwise PERMANOVA tests when the one-way tests resulted in $P < 0.05$) to compare communities among years of the tree mortality chronosequence. Both procedures were completed in PC-ORD using Bray-Curtis distance matrices (*McCune & Mefford, 2011*). Final stress for NMDS runs indicated reasonably well fit, 2 dimensional solutions for both aboveground and belowground arthropod communities, with stress interpretation following the suggestions of *Keough & Quinn (2002)* and *Clarke (1993)*. Prior to PERMANOVA, the data for both above and belowground communities were log

transformed and relativized to the maximum species abundance to account for differences in total abundances as described by *McCune & Medford (2002)*. Prior to all linear models, we verified that data were normally distributed via Shapiro–Wilk tests, verified homogeneity of variances via O'Brien tests and analysis of mean variances (ANOMV), and checked the distributions of residuals via Sharpio-Wilk tests and normal quantile plots. We then compared arthropod total abundance (log transformed to meet assumptions of normality), $\alpha$-diversity (sample-level species diversity calculated as the Shannon diversity index, H′), as well as soil chemical measures, and vegetation species richness and cover using one-way ANOVA followed by post hoc LSD means comparisons (Kruskal–Wallis test followed by Wilcoxon pairwise comparisons when assumptions of normality were not met).

We used null deviation analysis to further assess assembly processes structuring both above and belowground arthropod communities across the tree mortality chronosequence. Null deviation analyses used alone can be difficult to interpret and were employed here as a complimentary approach to linear modeling and Mantel tests for assessing the factors structuring arthropod communities; detailed description of null deviation methods and R code are available from *Chase & Myers (2011)* and *Tucker et al. (2015)*. In brief, the null deviation method assesses how observed $\beta$-diversity patterns deviate from communities randomly assembled *in silico* from the regional species pool. This approach disentangles the dissimilarity in structure across samples from dissimilarity driven by changes in $\alpha$ (local) and $\gamma$-(regional) diversity. We calculated null deviation as the relative difference of observed $\beta$-diversity from null modeled $\beta$-diversity—i.e., $(\beta_{obs} - \beta_{null})/\beta_{null}$, where $\beta$-diversity was measured as Sørenson-Czekanowski binary dissimilarity. For each sample, null modeled $\beta$-diversity was calculated from 10,000 randomly assembled communities. We compared null deviation values of aboveground and belowground communities via a permutation test that resamples from null deviation values generated by five unique null deviation simulations Permutation tests shuffle the labels of factors (i.e., aboveground vs belowground group labels are shuffled among null deviation output values) to compare the number of differences between factors that are more extreme than the difference with unshuffled factors (i.e., differences from aboveground null deviation value— belowground null deviation value vs. differences from the same calculation when the labels are shuffled) (*Yu, 2003*). The null hypothesis of the permutation test is that the mean null deviations for aboveground and belowground arthropod communities within years of the chronosequence are equal; the reported *P*-values indicate the likelihood that observed differences in null deviation among the communities is due to chance (i.e., smaller *P*-values indicate lower likelihood that the two groups differ by chance alone).

Following null modeling, we examined possible relationships of vegetation and soil properties (independent variables) with aboveground/belowground arthropod community structure (dependent variables) via Mantel tests. Mantel tests were completed using Sørenson distance matrices for arthropod communities and Euclidean distance matrices for environmental factors. We also examined possible relationships between vegetation and soil properties (independent variables) and arthropod abundance and diversity (dependent variables) via stepwise multiple regressions. Independent variables used in both Mantel tests and regression models included: soil moisture, pH, %C, %N, C:N, DOC, $NH_4^+$,

**Table 1** Results of one-way PERMANOVA tests of ground-dwelling arthropod community structure among years of a 5-year chronosequence of insect-induced tree mortality.

| Community | Source | *df* | SS | MSE | *F* | *P* |
|---|---|---|---|---|---|---|
| Aboveground | Year | 4 | 1.54 | 0.385 | 2.14 | 0.0002 |
| | Residual | 35 | 6.29 | 0.180 | | |
| | Total | 39 | 7.84 | | | |
| Belowground | Year | 4 | 1.68 | 0.420 | 1.51 | 0.0200 |
| | Residual | 25 | 6.94 | 0.277 | | |
| | Total | 29 | 8.62 | | | |

vegetation species richness total vegetation cover, forb cover, graminoid cover, tree cover, shrub cover, and rock cover. Best-fit multiple-regression models were selected via Bayesian information criterion (BIC) values, with the lowest BIC score indicating the model that explained the most variation in arthropod measures with the smallest number of factors to avoid over-fitting. Independent variables retained in regression models were examined for collinearity via correlation coefficients (i.e., collinear measures with $P > 0.05$ were avoided).

## RESULTS

### Arthropod community structure and tree mortality

We captured a total of 10,757 individual arthropods, representing 39 morphospecies (hereafter referred to as "species") collectively across all aboveground (23 spp., sampled via pitfall traps) and belowground samples (20 spp., sampled via modified Winkler extractors) with four species shared among both groups (Table S1 and Fig. S1). There was an average of 11 species in each aboveground sample across the chronosequence; with 14 of the 23 species found in all five years of the chronosequence. For belowground arthropods, there was an average of 5 species per sample, with 6 of the 20 belowground species found in all chronosequence years.

Aboveground arthropod species richness (displayed throughout as the mean ± 1 SE) did not significantly differ across years, with the lowest richness of 9.6 (±1.1) found three years after tree mortality and the highest richness of 11.4 (±2.5) found four years after tree mortality in the final year of the chronosequence. Tree mortality did significantly alter aboveground arthropod abundance ($F = 6.7$, $d.f. = 4, 35$, $P = 0.0004$; Fig. 1) and species diversity (H′), ($F = 8.3$, $d.f. = 4, 35$, $P < 0.0001$; Fig. 1). In the belowground arthropod community, tree mortality did not have a significant effect on either arthropod abundance or diversity (H′) ($P > 0.05$; Fig. 1). Despite the variable effects of tree mortality on abundance and diversity between above and belowground communities, tree mortality did cause significant shifts ($P < 0.05$) in community structure in both the aboveground and belowground arthropod communities. Changes in community structure were primarily driven by differences between communities of years 3 and 4 and those in years 1 and 2 for aboveground arthropods, and a difference between year 3 and year 0 (undisturbed) for both above and belowground communities (Fig. 2 and Table 1).

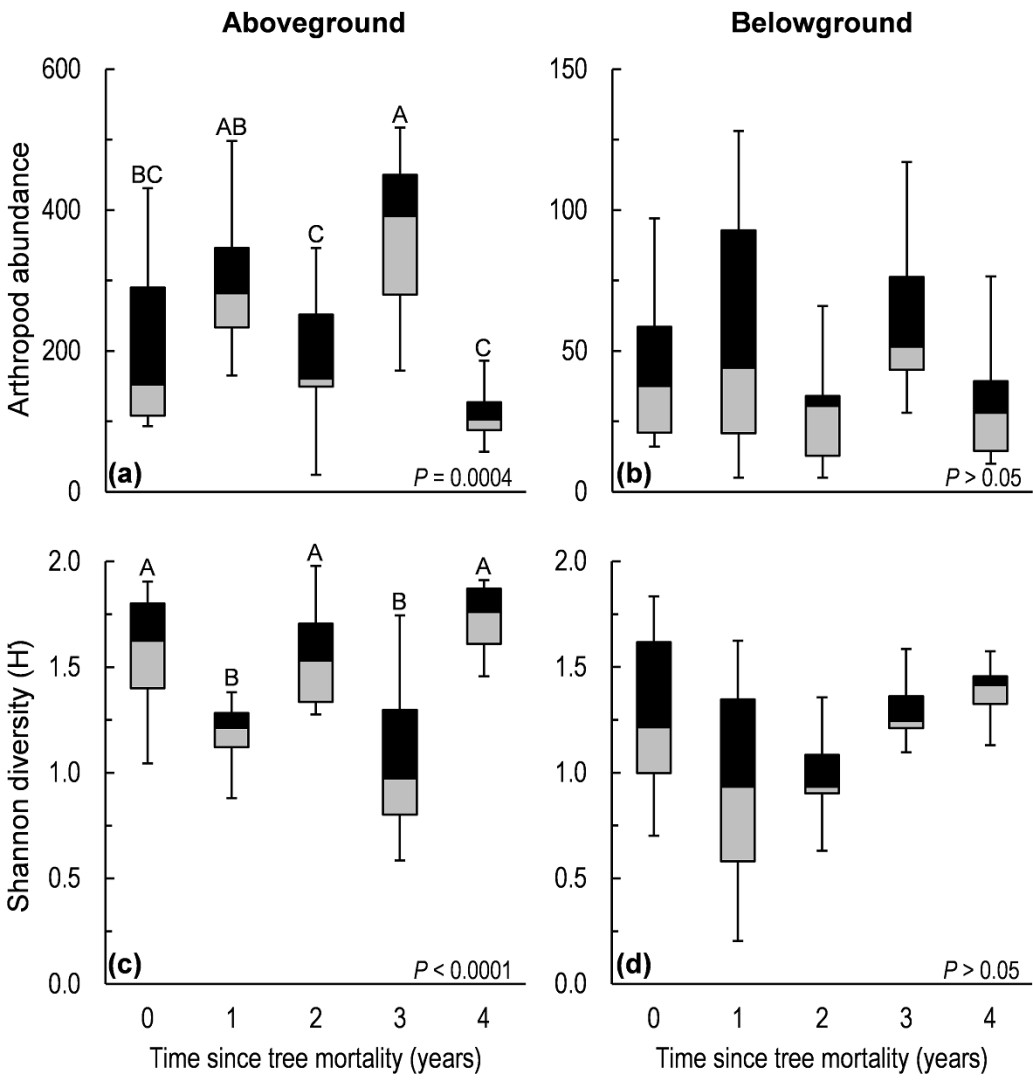

**Figure 1** **Abundance and Shannon diversity (H′) of aboveground and belowground arthropod communities sampled along a five-year chronosequence of insect-induced tree mortality.** Box and whisker plots show the median (center line), the 1st and 3rd quartiles (shaded boxes), and the 1.5 inter-quartile range or ∼97% of variation in the untransformed data (whisker bars). Boxes with different letters are significantly different ($P < 0.05$) via LSD means comparisons following one-way ANOVA.

## Community assembly processes

We assessed community assembly processes via the null deviation approach (*Chase & Myers, 2011*). This approach compares observed levels of $\beta$-diversity in field samples to the $\beta$-diversity of samples randomly assembled in computer simulations to produce an index that ranges from ±1 to 0, where values closer to ±1 indicate greater deviation from random (suggesting a stronger relative influence of deterministic assembly processes). Null deviation values suggested that tree mortality altered the relative influences of deterministic and stochastic assembly processes in both aboveground and belowground arthropod communities. Following tree mortality, null deviation values for belowground communities declined in absolute value across years 1 through 4 of the chronosequence reaching a low

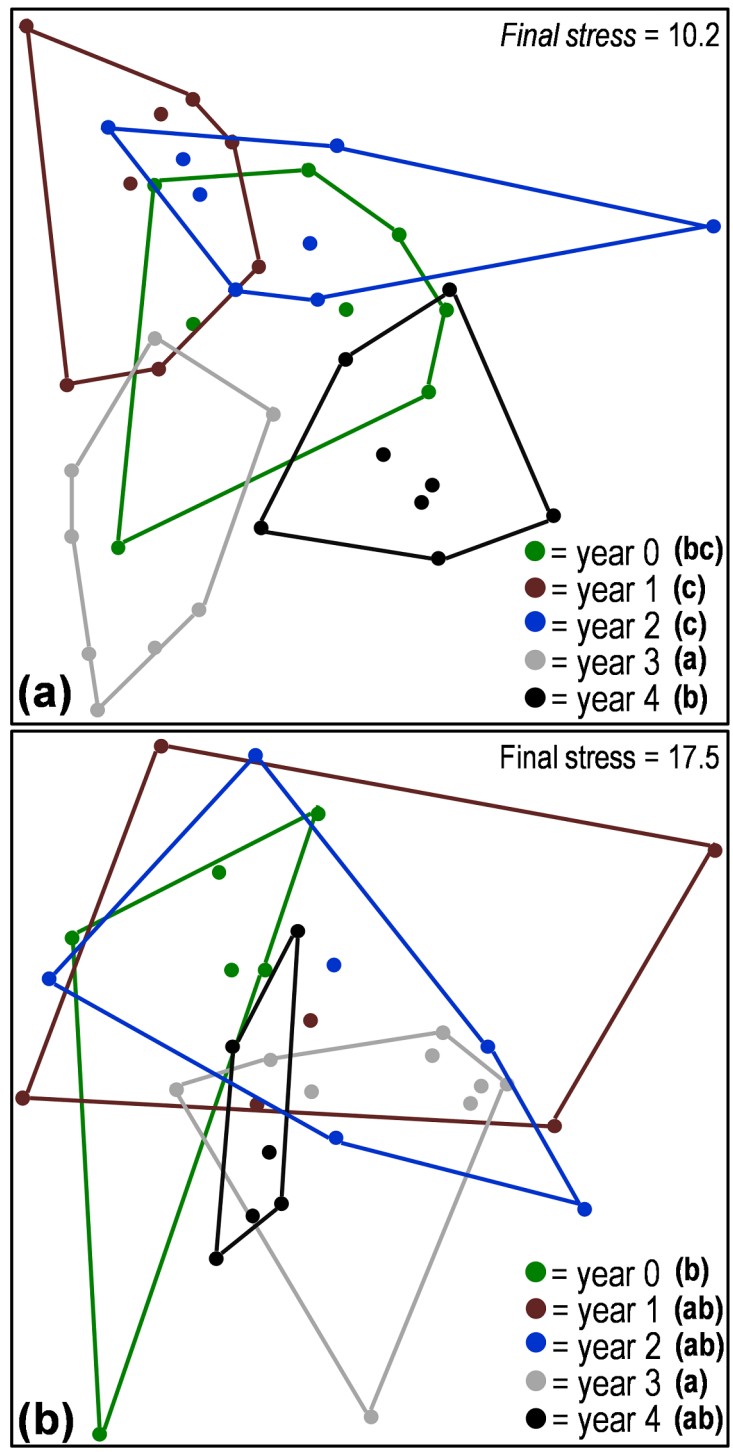

**Figure 2** **Non-metric multi-dimensional scaling (NMDS) ordination based on Bray–Curtis distances comparing the structure of aboveground (A) and belowground (B) arthropod communities from samples collected along a five-year chronosequence of insect-induced tree mortality.** Chronosequence years with different letters in the legend indicate communities that are significantly different (PERMANOVA $P < 0.05$).

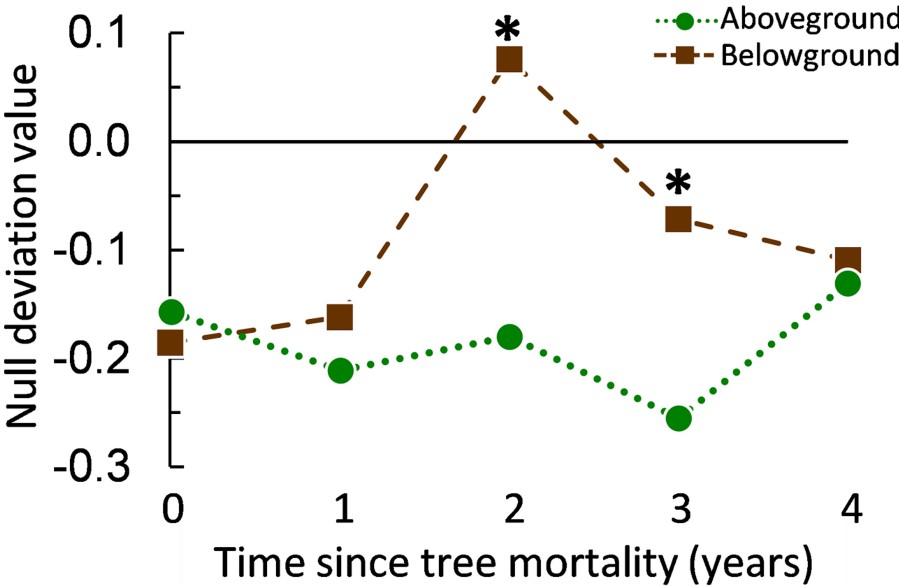

**Figure 3** **Null deviation values from aboveground and belowground arthropod communities sampled along a five-year chronosequence of insect-induced tree mortality.** Null deviation values close to zero indicate species compositions that deviate less from random suggesting a greater relative influence of stochastic processes on community assembly, larger values (negative or positive) indicate increasing deviation from random and suggest greater relative influence of deterministic processes. Null deviation values of above- and belowground communities significantly differ within chronosequence years 2 and 3 as indicated by an asterisk (*) above the higher symbol.

of |0.07|, indicating more stochastic assemblages (Fig. 3, Table S2). In contrast, deviation from randomly assembled communities increased slightly for aboveground communities in years 1 and 3 after tree mortality, suggesting a stronger and/or stable relative influence of deterministic processes on community assembly (Fig. 3). Permutation tests revealed a significant difference ($P < 0.05$) in the null deviation of aboveground and belowground communities for years 2 and 3 of the chronosequence (Fig. 3). However, an increase in stochastic influences in aboveground communities was apparent in the final year of the chronosequence (year 4; Fig. 3, Table S2) suggesting that above- and belowground communities may have returned to experiencing similar relative influences of different assembly processes.

## Associations of arthropod community structure and vegetation/soil properties

Tree mortality led to variation in soil chemical properties across the chronosequence (Table S2, see also *Ferrenberg et al., 2014*), and caused significant changes in understory vegetation cover ($F = 4.6$, $d.f. = 4, 35$, $P = 0.004$; Fig. 4) and vegetation species richness ($F = 4.8$, $d.f. = 4, 35$, $P = 0.004$; Fig. 4). Differences in relative cover of plant functional groups was also found across the chronosequence: forb cover increased seven-fold between year 0 and 3 (Table 2), and both graminoid and shrub cover increased by an order of magnitude or more between year 0 and 2 (Table 2).

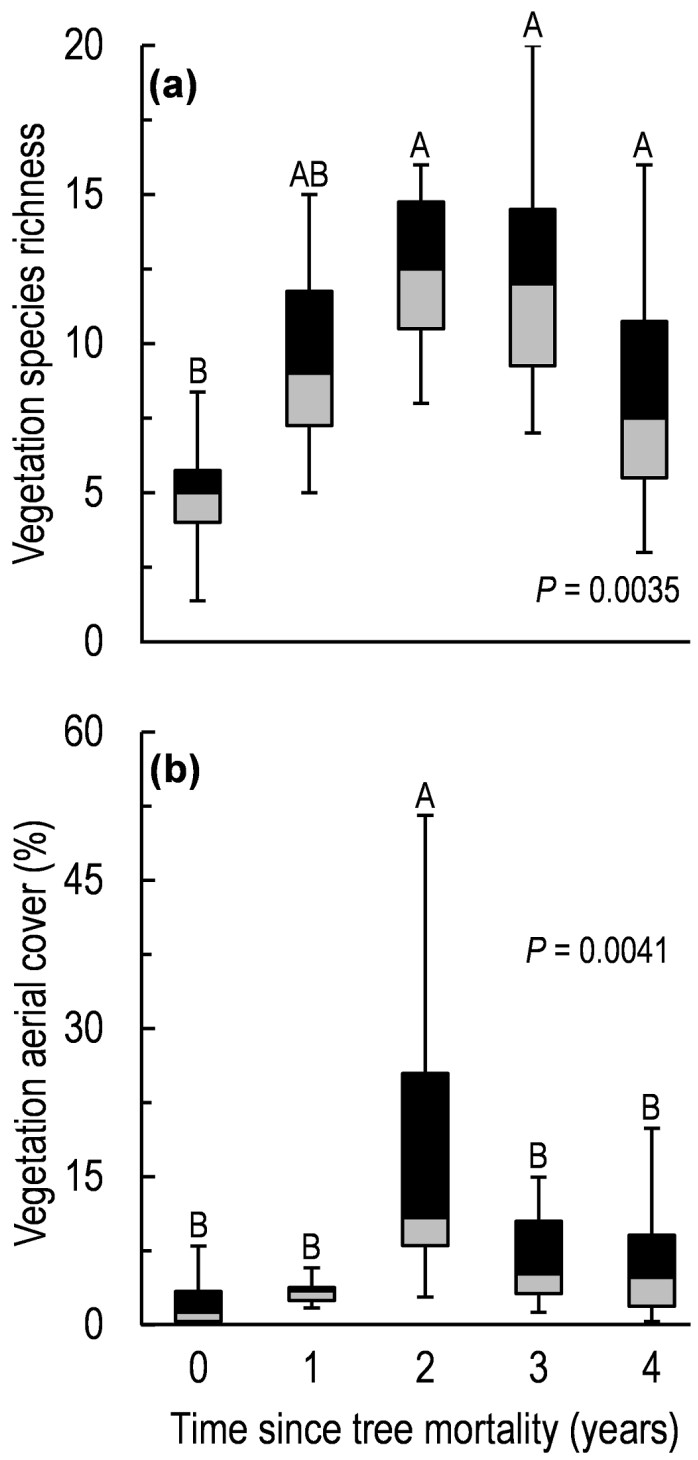

**Figure 4** **Vegetation species richness (A) and aerial cover (B) along a five-year chronosequence of insect-induced tree mortality.** Box and whisker plots show the median (center line), the 1st and 3rd quartiles (shaded boxes), and the 1.5 inter-quartile range or ∼97% of variation in the untransformed data (whisker bars). Boxes with different letters are significantly different ($P < 0.05$) via LSD means comparisons following one-way ANOVA.

**Table 2  Percent cover of vegetation types across a five-year chronosequence of insect-induced tree mortality.**

| Year | Forb | | Gramminoid | | Shrub | | Tree | |
|---|---|---|---|---|---|---|---|---|
| 0 | 1.5 | $(\pm 0.7)^b$ | 0.9 | $(\pm 0.4)^c$ | 3.4 | $(\pm 3.0)^b$ | 1.9 | $(\pm 1.0)$ |
| 1 | 3.0 | $(\pm 0.6)^{ab}$ | 1.5 | $(\pm 0.3)^b$ | 3.6 | $(\pm 1.3)^{ab}$ | 4.2 | $(\pm 2.9)$ |
| 2 | 7.2 | $(\pm 2.2)^a$ | 14.9 | $(\pm 9.5)^a$ | 33.9 | $(\pm 12.2)^a$ | 3.2 | $(\pm 1.7)$ |
| 3 | 10.5 | $(\pm 3.6)^a$ | 5.2 | $(\pm 1.3)^a$ | 4.7 | $(\pm 3.5)^b$ | 0.7 | $(\pm 0.5)$ |
| 4 | 7.9 | $(\pm 4.3)^{ab}$ | 1.8 | $(\pm 0.5)^{bc}$ | 11.3 | $(\pm 9.9)^{ab}$ | 1.5 | $(\pm 0.7)$ |
| *P-value* | <0.05 | | <0.001 | | <0.05 | | >0.05 | |

**Notes.**

Values are untransformed means ± 1 SE, *P*-value from one-way ANOVA (Kruskal–Wallis tests when assumptions of normality were not met). Means followed by different letters are significantly different ($P < 0.05$) based on LSD or Wilcoxon post-hoc comparisons.

Mantel tests revealed a significant association between aboveground arthropod communities and combined vegetation/soil properties ($r = 0.26$, $P = 0.005$), while belowground communities were not significantly associated with vegetation/soil properties ($r = 0.05$, $P > 0.05$). Partial Mantel tests further indicated that Aboveground arthropod communities were significantly associated with total vegetation cover ($r = 0.28$, $P = 0.016$), graminoid cover ($r = 0.34$, $P = 0.018$), shrub cover ($r = 0.20$, $P = 0.030$), and soil $NH_4^+$ concentration ($r = 0.16$, $P = 0.026$); while all other environmental variables were not significantly associated ($P > 0.05$) with the arthropod community. Stepwise multiple regression models identified significant relationships between vegetation/soil properties and aboveground arthropod abundance and diversity, but no significant relationships for belowground arthropods ($P < 0.05$). Specifically, aboveground arthropod abundance was significantly associated with total vegetation cover, vegetation species richness, and total soil carbon (%C); while arthropod species diversity (Shannon H′) was related to shrub and rock cover, and total soil carbon concentration (%C)

# DISCUSSION

We investigated the effects of tree mortality on the structure and assembly of arthropod communities (characterized at the level of morphospecies) along a five-year chronosequence of bark beetle-induced tree death in a subalpine conifer forest. Given the existence of substantial variation in the ecology and dispersal potential of aboveground versus belowground arthropods (*Blossey & Hunt-Joshi, 2003*; *De Deyn & Van der Putten, 2005*; *Joern & Laws, 2013*), we examined both communities separately with the goal of understanding whether the disturbance from tree mortality had contrasting effects on these different fractions of the ground-dwelling arthropod community. We found support for our first hypothesis that tree mortality caused a shift in arthropod community structure; a result that was true for both above and belowground arthropod assemblages (Fig. 2 and Table 1). However, tree mortality appeared to have a greater effect on the structure of the aboveground arthropod community than on the belowground, as evidenced by the changes in abundance and diversity in aboveground arthropods

but not in belowground arthropods (Fig. 1). We also observed changes in understory vegetation cover and vegetation species richness following tree mortality (Fig. 4 and Table 2), as well as variation in edaphic properties (Table S1). Yet despite changes in vegetation and soil properties, we found only mixed support for our second hypothesis that changes in the understory environment following tree mortality would lead to an increased influence of deterministic processes in the assembly of arthropod communities. Specifically, null deviation models (*Chase & Myers, 2011*) comparing the relative deviation of observed communities from communities randomly assembled *in silico* suggested that both aboveground and belowground communities experience a similar balance of assembly processes in undisturbed sites (Fig. 3). Yet following tree mortality, we observed a stable, stronger relative influences of deterministic processes in the assembly of aboveground communities than apparent for belowground communities which experienced a significant increase in the relative influence of stochastic assembly processes (Fig. 3).

A stronger influence of deterministic processes in structuring aboveground versus belowground communities is further supported by multiple regression models and Mantel tests of association. Specifically, multiple regression models found a significant relationship of both arthropod abundance and diversity to a mixture of vegetation and soil properties (Table 3). Also, in Mantel tests of association, the overall community structure (the combination of composition, diversity and abundance) of aboveground arthropods was significantly associated with various environmental factors including: total vegetation cover, graminoid (grass) cover, shrub cover and soil ammonium ($NH_4^+$) concentrations—a measure that often increases in the short term after tree mortality due to a decline in overall uptake (*Morehouse et al., 2008*; *Mikkelson et al., 2013*). At the same time, neither analysis found a link between belowground community structure and environmental factors we measured here, suggesting a weaker relationship to local environmental properties following tree mortality. However, it is possible that other environmental factors that were not sampled in our study have an influence on belowground arthropod community structure. At the same time, we utilized soil chemical measures collected several months before sampling arthropod communities. This time lag between sampling campaigns likely allowed for some change in soil chemical pools which could have reduced the apparent influence of soil properties on arthropod communities. Nevertheless, the model associations between arthropod abundance and diversity suggest that the return to a similar aboveground arthropod community in the final year of chronosequence (year 4) as found in the undisturbed (year 0) portion of our chronosequence (Fig. 2 and Table 3) is driven by arthropod responses to vegetation properties and soil carbon dynamics—factors that have similar dynamics to aboveground arthropod responses following tree mortality (Fig. 4, Table S1).

Bark beetle infestations have impacted enormous swaths of western North America, leaving billions of dead trees in their wake, often at higher elevations and latitudes than previously recorded due to rapidly warming temperatures (*Mitton & Ferrenberg, 2012*; *Mitton & Ferrenberg, 2014*). Tree mortality during recent epidemics has been linked to increased understory vegetation productivity (*Brown et al., 2010*); as well as changes in forest microclimate (*Wiedinmyer et al., 2012*; *Maness, Kushner & Fung, 2013*), soil

**Table 3** Best fit models relating vegetation cover and soil factors to total abundance and Shannon diversity (H′, α-diversity) of the aboveground arthropod community

| Response variable | Predictor variable[b] | F | P | adj. $R^2$ | BIC[a] |
|---|---|---|---|---|---|
| Arthropod abundance | Veg. cover | 16.33 | 0.0003 | | 80.1 |
| | Veg. species richness | 8.59 | 0.0058 | 0.40 | 76.8 |
| | Soil carbon (%) | 4.46 | 0.0416 | | 75.8 |
| Arthropod diversity (Shannon H′) | Shrub cover | 6.54 | 0.0149 | | 35.2 |
| | Soil carbon (%) | 6.48 | 0.0153 | 0.26 | 33.9 |
| | Rock cover | 3.78 | 0.0598 | | 33.7 |

Notes.

[a] Bayesian information criterion (BIC) cumulative values with the addition of the given line's predictor; in both cases, all three listed predictors were retained in the best fit model–i.e., the lowest BIC score of all models.

[b] Possible predictor variables included total vegetation cover, vegetation species richness, forb cover, graminoid cover, tree cover, shrub cover, and rock cover; along with soil moisture, C, N, C:N, DOC, $NH_4^+$, and pH. Variables retained in best fit models were screened for collinearity to avoid over-fitting models. Belowground arthropod measures were not significantly influenced by vegetation or soil properties.

hydrology (*Mikkelson et al., 2013*), and soil nutrient pools (*Morehouse et al., 2008*; *Griffin, Turner & Simard, 2011*; *Xiong et al., 2011*; *Griffin & Turner, 2012*). Thus, a shift in ground-dwelling arthropod community structure in response to tree mortality is not surprising given arthropod community sensitivity to changes in vegetation and litter cover from various forest disturbances, ranging from severe wildfires to relatively minor perturbations such as manipulations of coarse woody debris (*Ferrenberg et al., 2006*; *Moretti, Duelli & Obrist, 2006*; *Lessard et al., 2011*; *Ober & DeGroote, 2011*; *Armitage, Ho & Quigg, 2013*; *Arnan, Cerdá & Retana, 2015*; *Delph et al., 2014*; *Williams et al., 2014*; *Brunbjerg et al., 2015*). Additionally, the shift in arthropod community structure we found here joins recent reports indicating that bark beetle-induced tree mortality alters the structure of soil fungal communities (*Treu et al., 2014*; *Štursová et al., 2014*) and nematode community trophic composition (*Xiong et al., 2011*) of European and North American conifer forests, respectively. Considered collectively, the changes in arthropod communities and understory vegetation structure we found here, and the changes in nematode and fungal communities found in other forests would seem to indicate that tree mortality during insect epidemics can widely affect forest-understory biotic communities. However, our finding that surface dwelling arthropods are more strongly influenced by environmental properties than belowground arthropods suggests the presence of complicated aboveground-belowground linkages affecting responses in these systems (*De Deyn & Van der Putten, 2005*; *Bardgett & Wardle, 2010*).

Given changes in the forest understory environment, we initially expected that changes in arthropod community structure following tree mortality would be linked to niche dynamics. However, the structure of biotic communities can be shaped by either deterministic processes (often interchanged with 'niche-based processes') or stochastic processes (sometime conflated with 'neutral processes'), and an increasing amount of evidence indicates a simultaneous influence of both processes in arthropod and macroinvertebrate communities (*Hart, 1992*; *Thompson & Townsend, 2006*; *Chase, 2007*; *Chase et al., 2009*; *Ellwood, Manica & Foster, 2009*; *Rominger, Miller & Collins, 2009*;

*Lepori & Malmqvist, 2009*; *Barber & Marquis, 2011*; *Fišer, Blejec & Trontelj, 2012*; *Joern & Laws, 2013*; *Kitching, 2013*). The variation we found in strength of assembly processes across fractions of the arthropod community indicates that disturbance can either increase or decrease the ratio of deterministic to stochastic processes within a community (e.g., *Didham, Watts & Norton, 2005*; *Leibold & McPeek, 2006*; *Chase, 2007*; *Lepori & Malmqvist, 2009*). While this outcome seems to complicate the goal of understanding how disturbance impacts community assembly, the relationship between disturbance and assembly processes is likely dependent upon regional species diversity, species dispersal rates, and the spatial and temporal scale of disturbances—all of which are expected to vary across systems and taxonomic groups (*Cottenie, 2005*; *Reed et al., 2000*; *Mackey & Currie, 2001*; *Mouquet & Loreau, 2002*; *Chase, 2003*; *Tuomisto, Ruokolainen & Yli-Halla, 2003*; *Vanschoenwinkel et al., 2007*; *Rominger, Miller & Collins, 2009*; *Lepori & Malmqvist, 2009*; *Márquez & Kolasa, 2013*). The interaction of these variables, alongside the effects of disturbances, in moderating the balance of deterministic and stochastic assembly processes are all but certain to generate a range of context-dependent outcomes across studies. Nevertheless, in our study system, a combination of temporal gradients and influences of distributions and dispersal rates likely explain the contrasting influences of deterministic and stochastic processes for above and belowground arthropod communities. Specifically, dispersal limitations likely inhibit the rate of niche-tracking and species sorting by belowground arthropods, at the same time as stochastic dispersal and heterogeneous distributions (linked to ecological strategies and landscape legacy) influence community assembly in the short term following tree mortality. Given enough time for dispersal, biotic-interactions and environmental filtering would begin to influence belowground arthropods, thereby explaining the greater relative influence of deterministic processes in undisturbed sites of the chronosequence (Fig. 3). This scenario agrees with recent work in passively dispersed soil microbial communities where disturbance caused an initial increase in stochastic influences on community assembly— likely due to a decline in species abundance at the same time as stochastic dispersal affected recolonization—with a shift toward deterministic influences over time as species diversity and abundance increased, leading to more biotic interactions and filtering (*Ferrenberg et al., 2013*; *Nemergut et al., 2013*). Meanwhile, aboveground arthropods, often being larger and more capable of rapid dispersal into suitable habitats than belowground arthropods, were more likely to experience biotic interactions and species sorting over the spatial and temporal scale of tree mortality in this forested system. Yet if these communities reach an equilibrium, stochastic processes could eventually exert greater levels of influence at larger spatial and temporal scales—possibly explaining the apparent increase in stochastic influences in aboveground communities in the final year of the chronosequence. This hypothesized scenario for aboveground communities is further supported both by linear (multiple regression) and permutation models (Mantel correlation) used here, and also by studies in other arthropod and macro-invertebrate dominated systems where disturbance increased deterministic processes via environmental filtering, with an eventual shift toward greater influence of stochastic processes over time (*Chase, 2003*; *Chase, 2007*; *Lepori & Malmqvist, 2009*).

## CONCLUSIONS

Forest disturbances due to insect epidemics are historically natural events that have increased in frequency due to warming climate and other global and regional factors (*Mitton & Ferrenberg, 2012*; *Ferrenberg, Kane & Mitton, 2014*). Understanding how biotic communities respond to increasing rates of forest disturbance might offer insightful tests of ecological theory, while also informing forest management strategies for dealing with large-scale tree mortality (*Ferrenberg, 2016*). We found tree mortality during a bark beetle infestation altered the structure of aboveground and belowground arthropod communities. Null deviation models suggested that these different fractions of the arthropod community experience different relative influences of assembly processes following disturbance: with aboveground arthropod communities more influenced by deterministic processes and belowground communities by stochastic. Likewise, aboveground arthropod community structure was linked to vegetation and soil properties, while the belowground community had no clear links to environmental characteristics. An important next step will be determining if arthropod communities assembled via divergent processes have variable influences on ecosystem processes and functioning. One possibility is that stochastically assembled communities have less direct links to ecosystem processes, or perhaps less predictable influences than do deterministically assembled communities (*Ferrenberg et al., 2013*; *Ferrenberg et al., 2014*; *Nemergut et al., 2013*; *Knelman & Nemergut, 2014*). This scenario might help to resolve the enigma of why ground-dwelling arthropod assemblages influence ecosystem processes in some systems (*Seastedt & Crossley, 1984*; *González & Seastedt, 2001*; *Bradford et al., 2002*; *Vasconcelos & Laurance, 2005*; *Finér et al., 2013*), but not in others (*Seastedt, 1984*; *Hättenschwiler, Tiunov & Scheu, 2005*).

## ACKNOWLEDGEMENTS

We thank the University of Colorado's Biological Sciences Initiative, the University of Colorado's Mountain Research Station, and Jeffry Mitton for logistical support. We are indebted to the late Diana Nemergut for her insights and support throughout this research. Our manuscript was improved by the thoughtful comments of Manu Saunders and two anonymous reviewers.

### Funding

This work was supported by an award from the University of Colorado's Museum of Natural History to Scott Ferrenberg, and by the University of Colorado's Biological Sciences Initiative (BURST and UROP) support of Alexander Martinez. The funders had no role in study design, data collection and analysis, decision to publish, or preparation of the manuscript.

### Grant Disclosures

The following grant information was disclosed by the authors:

University of Colorado's Museum of Natural History.

University of Colorado's Biological Sciences Initiative (BURST and UROP).

### Competing Interests

The authors declare there are no competing interests.

### Author Contributions

- Scott Ferrenberg conceived and designed the experiments, performed the experiments, analyzed the data, contributed reagents/materials/analysis tools, wrote the paper, prepared figures and/or tables, reviewed drafts of the paper.
- Alexander S. Martinez and Akasha M. Faist performed the experiments, wrote the paper, reviewed drafts of the paper.

### Data Availability

Ferrenberg, Scott; akasha.faist@colorado.edu; mart1139@purdue.edu (2016): Ferrenberg_etal._arthropod data.xlsx. Figshare:

https://dx.doi.org/10.6084/m9.figshare.2765491.v1.

### Supplemental Information

Supplemental information for this article can be found online at http://dx.doi.org/10.7717/peerj.2545#supplemental-information.

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
