# Peer review of "Aboveground and belowground arthropods experience different relative influences of stochastic versus deterministic community assembly processes following disturbance"

_PeerJ, doi:10.7717/peerj.2545_

## Round 0.1 · original submission · Major Revisions

The three reviewers had widely divergent opinions. Two had a lot of problems with the paper and one thought it was fine with minor revisions. I also liked the paper and I think it is an important contribution. I am not put off by reviewer 1 saying that others have documented changes in soil fauna after disturbance. That almost goes without saying, so you can downplay that in the revision. However, for the paper to be acceptable you will have to address all of the review comments. Given the detailed problems provided by reviewers 1 and 2 It may be easier for you to try another journal.

Reviewer 1 ·

Basic reporting

First, the question “does tree mortality alter communities?” is not novel. There are many cases where the impacts of tree mortality (including insect-induced tree mortality) were documented on local invertebrate communities (see Adkins and Rieske, 2013, For. Ecol. Manage. 295: 126-135; Ingwell et al., 2012, Northeast. Nat. 19: 541-558; Delph et al., 2014, West. North Amer. Nat. 74: 162-184; for example, and there are many others). As such, the first paper hypothesis is, at best, confirming things we generally already know and accept. Yes, this is being investigated in a new tree type (Pinus flexilis) but the concept is the same as we see for any tree species that is impacted by insects (e.g. hemlock, ash, or any number of pines).

The same can be said for the second hypothesis, that changes in communities result from deterministic factors (e.g. changing vegetation, soil properties). Again, this is rather well known, especially in the literature dealing with natural disturbances (there are several good papers on this topic from the Luquillo Experimental Forest in Puerto Rico, especially the Canopy Trimming Experiment: http://luq.lternet.edu/research/project/canopy-trimming-experiment-cte). Those researchers have shown multiple times that ground arthropods aren’t impacted by the hurricane per se, but by the change in ground-level environment.

Experimental design

A major issue of this study, in my opinion, is the experimental design. The way I read it, each tree is a plot (5 chrono years * 8 trees = 40 focal trees). Traps in a plot were >15 m apart, and traps in different plots were at least 20 m apart. It is completely unclear how far apart different chronosequence areas were. Regardless, these distances are much too close to be considered independent samples, and this point is a huge issue for this study. Carabid beetles, for instance, can travel hundreds of meters in a single night (e.g. Negro et al., 2008, Eur. J. Entomol. 105: 105-112) and many travel 25m or more during the day (Negro et al. and Martay et al., 2014, Rest. Ecol. 22: 590-597; Allema et al., 2015, Bull. Entomol. Res. 105: 234-244). I also see the authors captured centipedes and ants, two groups known to cover a large area when searching for food. Ants, in particular, are tricky to sample with pitfalls as captures can be biased based on proximity to a nest.

The authors only captured 39 different taxa (FYI, Scolytinae is a Subfamily in the Family Curculionidae), of which several are very easily identifiable, with widely available taxonomic keys. The fact that they only identified their captures to morphospecies is disappointing and takes away from the knowledge that could be gained in this study. Were experts contacted to help with specimen identifications? If not, why? Further, the authors only sampled twice (June and August) in one calendar year. Few in the invertebrate community world would agree that this level of sampling is adequate to describe a community. What the authors sampled is a snapshot in time – temporal variation is immensely important in invertebrate population dynamics. This sampling method is likely adequate for soil microbes, as referenced in the 2014 publication by the authors, but for organisms that are mobile, I doubt this set-up really gives any relevant information, at least in its current state.

Why were June and August arthropod samples pooled together? Some justification of this is necessary.

I think the mobility of the aboveground arthropods makes conclusions with soil chemistry/characteristics spurious at best. Again, many of these critters travel a long ways on a daily basis, and can fly when they choose to do so.

Overall, I just don’t think the sampling design was adequate to answer the questions posed. Unfortunately, unless I’m completely wrong on the sampling design, I don’t think this set-up will work for mobile arthropods.

Validity of the findings

I do not think they are valid, please see Experimental Design comments.

Reviewer 2 ·

Basic reporting

I didn't see any raw data (Table S1 is not raw data), but I believe the manuscript meets other PeerJ 'basic reporting' criteria.

Experimental design

A central theme of the manuscript compares the response of aboveground communities to the response of belowground communities to tree mortality. The experimental design is not balanced, however

The null modeling approach is not clear enough. The authors need to provide all details of their approach, including specific algorithms and assumptions. As presented, if the raw data were provided, I would not know how to repeat their analyses. One specific point is the definition of the regional species pool. The authors note (line 228) that the regional pool was all arthropods found among samples. I think they are saying they combined belowground and aboveground taxa into one regional pool. If that's the case, the authors could say that more directly. In addition, a justification for that choice is warranted (instead of using separate regional pools for belowground and aboveground...note that a comparison among regional pool definitions can be very informative, I would suggest trying it out).

Also related to the null modeling, a justification for using the Sorensen metric is warranted. The authors have abundance information, which they use in other analyses, but they throw it away during the null modeling work. I would strongly recommend keeping that information. If the authors disagree, they should provide a strong justification for using only presence/absence data in the null modeling, but abundance information elsewhere.

On line 233 the authors note that gamma diversity was the entire species pool. I am not sure why this is noted because gamma diversity of the entire pool is not used in the calculation of Sorensen. Maybe I am missing something? Either way, the authors need to fully detail their null modeling method.

Also unclear is the mention (line 237) of using '5 unique simulations'. I don't understand what the authors are referring to here, please clarify.

On line 243 the authors say they used regression to look at pairwise dissimilarity. I don't understand that. I wonder if that what they really meant? Please check and clarify.

On line 245 there is no mention of the C:N ratio. Is there a reason that wasn't used?

Line 250: How was collinearity checked? What criteria were used? What was done when collinearity was found?

Validity of the findings

The authors state (line 268) that tree mortality significantly altered abundance and diversity in the aboveground fraction. Unfortunately, the supporting results (Figure 1) are not convincing. For abundance, only year 3 differs from year 0. It looks more like abundance is just bouncing around among the groups with no apparent trends. A very similar pattern is seen for diversity. In short, I don’t believe the authors inference is supported by the data.

Line 274-276 notes a pattern related to year 5,which doesn’t exist. Looking at the data shown in Figure 2, its clear that year 3 is again showing a significant difference (this time for community composition). A key aspect of this result is that only year 3 is different from year 0. I am therefore again not convinced that the data support the inference (in this case that mortality causes shifts in community composition). There appears to be something else in the system that is driving composition/abundance/diversity.

Regarding the null modeling results, it is important to note that all the null model deviations are non-significant, which indicates the predominance of stochastic processes. The authors do not mention this. I feel it is important that they discuss it.
When the null modeling results are presented the belowground results are presented first even though the aboveground results are the top panel in Figure 3. This caused me confusion for awhile until I realized the order was flipped. That would be an easy fix.

A major conclusion of the work is a difference in assembly processes between the aboveground and belowground components. In my opinion, Figure 3 shows small variations in null model deviations that are all well within the realm of stochasticity. If the authors plotted the data on the full scale, ranging from -1 to +1 the patterns would not look like much. I will try to upload the resulting figure, using data from Table S2. In addition, I did a simple t-test to compare the null model deviations between the two components and the p-value is > 0.2. There just isn’t enough data to come away with a strong conclusion. In other words, there may be some difference in reality between the aboveground and belowground, but I don’t think the data provide clear evidence of that difference and if there is a difference, it represents relatively small variations within a stochastic system.

Given that, I would strongly encourage the authors to use the abundance information in the null modeling. They may find that gives a much clearer signal of deterministic processes and a clearer difference between the two components.

For the vegetation response, I am afraid I have similar concerns. The vegetation species richness shows a nice pattern. The vegetation cover does not, however. The pattern show in the bottom panel of Figure 4 looks like there were a couple outlier plots in year two. My interpretation is that mortality didn’t impact cover in a systematic way, and that there is inherent variation in the system that is driven by some other feature.

Line 304: I can’t tell what was included in this Mantel test, so it’s difficult to gauge the associated inferences.

For the model selection, it would be useful to provide a table of model parameters and statistics. It would be even better to include those statistics for all models that were within about 2 BIC units so the reader can decide how much stock to put into inferences based on the best model.

Still not understanding the dissimilarity noted on line 308. It is not mentioned in the follow-on results.

And, there’s no mention of the collinearity tests that were promised in the Methods section.

I must say I had a very hard time with the Discussion. The authors have a neat sampling design and some tantalizing results, but nothing very definitive at this point. Yet, inferences and implications throughout the Discussion are written in a way that suggests the authors found definitive and major differences through time and across the compartments, and that these differences have serious implications. As summarized above, I don’t see any major differences and I don’t see any strong evidence of deterministic processes. At minimum, I recommend a very significant reduction in the length and speculation of the Discussion. Much better would be reconsidering the results and associated inferences, which would lead to a very different Discussion section.

Annotated reviews are not available for download in order to protect the identity of reviewers who chose to remain anonymous.

·

Basic reporting

The manuscript is well-written and structured and is easy to follow. Literature is well-referenced. Figures & tables are clear and captions provide sufficient information for interpretation.

Experimental design

In general, the study appears to be well-designed and Methods are described clearly.

Line 189: The authors say that chemical properties were measured in spring, while arthropod samples were collected in summer. Would seasonal differences in soil chemistry affect the relationships you looked for between arthropods & soil properties?

Validity of the findings

Analysis is appropriate for the data and study design. Conclusions are discussed clearly in light of research questions and results. Some acknowledgement of study limitations are discussed.

Line 347-8: see comment above about the time lag between soil chemistry & arthropod sampling. In your analysis you have essentially looked for associations between summer arthropod communities & spring soil chemistry, so this could explain the weaker relationships? Some clarification could help here.

---

## Round 0.2 · Major Revisions

Ok this paper presents a conundrum. Reviewers continue to be widely divergent in opinions. I still like the paper but reviewer number 1 emphatically does not. His issue predominately is that the sampling design does not support the conclusions - "even with a reference about the trap distance." Referee 2 has substantive responses to your revision that I want you to address. Reviewer 3 is fine with the revision. So, give me a counter to Reviewer 1 (see annotations on the draft) and respond carefully to Reviewer 2 in a rebuttal letter, give me another revision based upon those comments and I will make a decision on the paper.

Reviewer 1 ·

Basic reporting

See attached document.

Experimental design

See attached document.

Validity of the findings

See attached document.

Additional comments

See attached document.

Annotated reviews are not available for download in order to protect the identity of reviewers who chose to remain anonymous.

Reviewer 2 ·

Basic reporting

Please note that the authors primarily responded to my comments with conceptual arguments in their response letter. I have elected to respond back using a format that indicates my comments from the first round of review, the authors' response, and my follow-on comments from the second round of review.

First round: I didn't see any raw data (Table S1 is not raw data), but I believe the manuscript meets other PeerJ 'basic reporting' criteria.

Response: Raw data were/remain available via the supplied FigShare link.

Second round: I see chemistry data on figshare, nothing else. Maybe I am just missing something.

Experimental design

First round: A central theme of the manuscript compares the response of aboveground communities to the response of belowground communities to tree mortality. The experimental design is not balanced, however

Response: We acknowledge that chronosequence year 4 had one less replicate than the other years due to an animal removing both pitfall traps at that site—a fact we failed to explain originally and have corrected. Aside from that missing sample, we had the same number of replicates for all chronosequence categories.
We did have less total plots included in the belowground because these communities were sampled from soil/litter collections via modified Winkler funnels (of which we had a limited number) on lab benches. However, it’s important to note that we do not make direct comparison between above and belowground communities during linear modeling or Mantel testing and instead offer up the null deviation approach, a metric that is computed in a fashion akin to an effect size, to allow readers to compare the assembly of the communities.

Second round: I understand there are practical reasons for having fewer plots in the belowground communities, but whatever the reason, there are differences. The point of the paper is comparing belowground to aboveground, but differences in number of plots makes the comparison unclear/uncertain. Why not just test out whether this has an effect by decreasing the number of aboveground plots and re-running the analyses. If everything stands, then it’s not an issue. If something changes, then it’s an issue. This would be easy and informative.

##

First round: The null modeling approach is not clear enough. The authors need to provide all details of their approach, including specific algorithms and assumptions. As presented, if the raw data were provided, I would not know how to repeat their analyses. One specific point is the definition of the regional species pool. The authors note (line 228) that the regional pool was all arthropods found among samples. I think they are saying they combined belowground and aboveground taxa into one regional pool. If that's the case, the authors could say that more directly. In addition, a justification for that choice is warranted (instead of using separate regional pools for belowground and aboveground...note that a comparison among regional pool definitions can be very informative, I would suggest trying it out).

Response: The null models were calculated on separate species pools for the above- and below-ground communities as you recommended—a point that was unclear in the original paper. We have expanded the description of the null model calculations. The references included for the null models included Chase et al. 2011, where the method was first described and which also offers supplemental R code for repeating the method, and Tucker et al. 2015 where the method is further explored and additional R code is provided. The availability of these resources in now stated.

Second round: Okay, thank you.

##

First round: Also related to the null modeling, a justification for using the Sorensen metric is warranted. The authors have abundance information, which they use in other analyses, but they throw it away during the null modeling work. I would strongly recommend keeping that information. If the authors disagree, they should provide a strong justification for using only presence/absence data in the null modeling, but abundance information elsewhere.

Response: We calculated the null models using a binary distance metric for reasons described in the extended mathematical paper by Chase et al. 2011. In a nutshell, the binary metric reduces the length of time needed to compute the null model (which can be quite substantial) and helps to disentangle the effect of alpha diversity on beta diversity which is a longstanding problem in community ecology. However, Tucker et al. 2015, found that other distance measures can be used in the null modeling framework under some circumstances, thus, we have directed readers to this paper for explanations of alternate calculations and caveats related to the modeling approach in general.

Second round: From this response I think the authors primary reason for not using the abundance information is because it will take longer on the computer. That’s not a great reason for throwing out the abundance data. I still encourage them to use it. Referring the reader to Tucker et al. isn’t going to tell the reader what the results look like when abundance information is included.

##

First round: On line 233 the authors note that gamma diversity was the entire species pool. I am not sure why this is noted because gamma diversity of the entire pool is not used in the calculation of Sorensen. Maybe I am missing something? Either way, the authors need to fully detail their null modeling method.

Response: Null beta diversity within the given null modelling equation is calculated by random sampling from the total community (gamma) to generate simulated samples (alpha). In short, the simulation is randomly generating communities from our observed samples as commonly done in null modelling. This method controls for changes in alpha and gamma diversity which are strong influences on beta diversity (i.e., two samples with different numbers of total species have a larger likelihood of being different simply because one has more individuals than the other). For example, the above and belowground communities have different gamma diversity levels, but the simulation approach generates a null deviation value that is independent of gamma diversity such that the value is akin to an effect size. This approach allows us to compare outcomes across different communities.
To avoid confusion, we have deleted the reference to gamma diversity and instead have included citations for Chase et al. 2011 and Tucker et al. 2015 with a note that the mathematical basis for these models is described in detail in these two references. We understand the need to explain our methodology, and as null models are now widely utilized in community ecology, we feel that directing readers to the mathematical detail supplied by these peer-reviewed methodological papers and their available R code is the most efficient way of transferring this information following our simplified description of the method.

Second round: Referring readers to those paper for details is a good idea, thank you. But, check the references as they are different than the Chase et al. referred to here.

##

First round: Also unclear is the mention (line 237) of using '5 unique simulations'. I don't understand what the authors are referring to here, please clarify.

Response: We ran each simulation (a group of 10000 permutation is considered one simulation) 5 times in order to verify stability and variation of null deviation values. In relation to your comment below, we have now conducted a permutation resampling of these simulations to compare the null deviation values among above-and belowground communities within each year of the chronosequence.

Second round: Interesting, but see comments below.

##

First round: On line 243 the authors say they used regression to look at pairwise dissimilarity. I don't understand that. I wonder if that what they really meant? Please check and clarify.

Response: The reference to dissimilarity was a mistake and has been deleted since we now reserve statistical comparisons of beta diversity to null modelling. The full sentence now reads: “We also examined possible relationships between vegetation and soil properties (independent variables) and arthropod abundance and diversity (dependent variables) via stepwise multiple regressions.”

Second round: Okay, thank you.

##

First round: On line 245 there is no mention of the C:N ratio. Is there a reason that wasn't used?

Response: We did not include C:N ratios because of collinearity among this ratio and the measures of C and N individually (i.e., C:N is a mathematical function of both C and N such that the ratio is linearly related to both variables). Including C:N in best fit models alongside C and N leads to over-fitting due to this collinearity. Thus, to avoid violating an assumption of multiple regression, we opted to keep each raw element pool as an individual variable.

Second round: Okay. This choice needs to be directly stated.

##

First round: Line 250: How was collinearity checked? What criteria were used? What was done when collinearity was found?

Response: We assessed collinearity of independent variables via correlation coefficients, which is now explained in the text. When collinearity was encountered (a situation isolated to vegetation cover measures) we retained the veg cover variable that led to the best overall fit model assessed via BIC scores.

Second round: Okay, thank you.

Validity of the findings

First round: The authors state (line 268) that tree mortality significantly altered abundance and diversity in the aboveground fraction. Unfortunately, the supporting results (Figure 1) are not convincing. For abundance, only year 3 differs from year 0. It looks more like abundance is just bouncing around among the groups with no apparent trends. A very similar pattern is seen for diversity. In short, I don’t believe the authors inference is supported by the data.

Response: We are confused by this comment, specifically the characterization that only year 3 differs from year 0. The figure actually shows that aboveground abundance in years 1 and 3 is significantly greater than years 2 and 4; while aboveground diversity in years 1 and 3 is significantly lower than in years 0, 2, and 4.

Second round: What I meant was that compared to year 0, only year 3 shows a significant difference in abundance. Note that I also said that it looks like abundance and diversity are just bouncing around without any systematic trends. That is definitely the case. Most importantly, I am evaluating whether the authors’ statements are backed up by the data. The authors state “Tree mortality did significantly alter aboveground arthropod abundance (F = 6.7, d.f. = 4, 35, P = 0.0004; Figure 1) and species diversity (H′), (F = 275 8.3, d.f. = 4, 35, P < 0.0001; Figure 1).” In my view, a clear sign of mortality impacting abundance or diversity would be a systematic change relative to year 0. By systematic change I mean a directional trend (consistently up/down), a step increase/decrease that is maintained through time, or maybe a unimodal function (e.g., a decrease for a couple years followed by tracking back toward the initial condition). The authors find nothing of the sort. They see abundance and diversity going up and down year after year with no trends. In other words, the trees are dead, but the metrics fluctuate up and down through time. There is therefore no evidence that tree mortality caused shifts in abundance or diversity.


##


First round: Line 274-276 notes a pattern related to year 5,which doesn’t exist. Looking at the data shown in Figure 2, its clear that year 3 is again showing a significant difference (this time for community composition). A key aspect of this result is that only year 3 is different from year 0. I am therefore again not convinced that the data support the inference (in this case that mortality causes shifts in community composition). There appears to be something else in the system that is driving composition/abundance/diversity.

Response: The reference to year 5 was a typo and should have read “year 4”. This has been corrected.
Again, we are confused by this interpretation of our results: we agree that aboveground arthropod community composition in year 3 differs from the undisturbed control (year 0). Yet the figure also illustrates that years 3 and 4 not only differ significantly from each other, but also from years 1 and 2. Perhaps the lettering in Figure 2 or description in the text was unclear? To this end, we have improved the description on lines 274-276.
It would appear that your interpretation is that communities in disturbed plots can only be compared to communities from undisturbed plots and not across years since disturbance. We disagree with that approach because this is a successional gradient where communities are sampled from 4 different time periods since disturbance and from undisturbed plots. This type of substitution of ‘space for time’ along ecological chronosequences is quite common and the aim is to understand how arthropod communities change both in response to disturbance and over time as communities respond to environmental change. The power of this design is in the ability to compare communities from different points in succession under similar abiotic conditions allow us to disentangle variation in community assembly from variation that would be caused by colder vs. warmer years, difference in precipitation, etc.

Second round: I stared at this for a while and I am coming around to the authors’ interpretation, although I still feel the evidence is relatively weak. Related to my comments above, a more systematic change in composition would be more convincing. I also am having a hard time getting away from the fact that going from a live tree to a dead tree did not have an immediate effect. One may say that there is a time lag following tree death. That would be fair, but then why would year 4 track back towards year 0? (note that year 4 is not different than year 0). There is some hint in the Figure 2 that tree mortality is a driver of shifts in composition, but it is tenuous. I would strongly recommend the authors provide appropriate caveats and also directly acknowledge and discuss that there is not a systematic trajectory in community composition and specifically discuss why year 4 tracks back to year 0 community composition.

##

First round: Regarding the null modeling results, it is important to note that all the null model deviations are non-significant, which indicates the predominance of stochastic processes. The authors do not mention this. I feel it is important that they discuss it.
When the null modeling results are presented the belowground results are presented first even though the aboveground results are the top panel in Figure 3. This caused me confusion for awhile until I realized the order was flipped. That would be an easy fix.
A major conclusion of the work is a difference in assembly processes between the aboveground and belowground components. In my opinion, Figure 3 shows small variations in null model deviations that are all well within the realm of stochasticity. If the authors plotted the data on the full scale, ranging from -1 to +1 the patterns would not look like much. I will try to upload the resulting figure, using data from Table S2. In addition, I did a simple t-test to compare the null model deviations between the two components and the p-value is > 0.2. There just isn’t enough data to come away with a strong conclusion. In other words, there may be some difference in reality between the aboveground and belowground, but I don’t think the data provide clear evidence of that difference and if there is a difference, it represents relatively small variations within a stochastic system.
Given that, I would strongly encourage the authors to use the abundance information in the null modeling. They may find that gives a much clearer signal of deterministic processes and a clearer difference between the two components.

Response: A t-test requires that samples be independent; beta diversity is by definition not an independent measure, but rather a measure of species turnover that is dependent upon comparisons among samples. Comparing beta diversity across treatments or categories, our chronosequence years for example, cannot be done with a distribution based statistic that assumes independence. A common way to compare beta diversity is using a permutation test which we’ve now employed and noted that above- and belowground null deviation values significantly differ in years 2 and 3. We were initially hesitant about including this test and leading readers to over-interpret the model results. Instead, the null deviations were offered as a supplemental analysis to the Mantel tests and linear regressions to give an index of stochasticity (randomness) for each chronosequence year. However, keeping the number of simulations low reduces the influence of simulation numbers on statistical power. This analysis now verifies that null deviation values do significantly differ in the middle years of the chronosequence. Including just one more simulation run would also result in year 1 being significantly different, but again, for reasons related to statistical power vs. interpretation we have stuck with five simulation runs.
Our interpretation of how each community is assembled is based on the entirely of our analyses. For example, we found aboveground community structure is linked to environmental measures via a Mantel test and a multiple regression using diversity measures. We did not find similar links in the belowground community. These findings suggest deterministic processes linked to environmental filters are stronger aboveground. This suggestion is further supported by the null model which indicates a larger overall null deviation in the aboveground community than in the belowground community following disturbance (both communities start with a similar null deviation). It is through the entirety of the analyses that we hope readers will interpret our results as well.

Second round: I understand that it’s difficult to compare null model results. The null model should provide a degree of independence, so while a t-test is not formally appropriate, I do believe it provides a sense of what’s going on. I think I understand the permutations that were done although I don’t find that very helpful either as the distributions of null deviations are likely very tight because 10000 randomizations are being used in the null modeling (i.e. if errors bars were included in Figure 3, they would be extremely small). It’s clear enough without the permutations that in years 2 and 3 there are differences between aboveground and belowground in the null deviations.
I must note that in their response the authors did not directly address my concern about the null deviations all being non-significant. I couldn’t upload the figure with the appropriate y-axis scale, but if the authors take a look at the null deviations with the y-axis from -1 to 1, it is quite apparent that the null deviations are well within the stochastic realm.
I can appreciate looking at the data holistically, and when the authors do that they should explicitly discuss the fact that their null model results indicate a stochastic system. Maybe that’s telling us something important or maybe that is due to an artefact/limitation of the null model. Along those lines, I will note again that I suggested to the authors to use abundance information in the null model to evaluate whether there is a clearer deterministic signal when that information is included. And I will further highlight that the authors are making lots of statistical comparisons between groups (aboveground and belowground) that differ in sample sizes and thus differ in statistical power.

##

First round: For the vegetation response, I am afraid I have similar concerns. The vegetation species richness shows a nice pattern. The vegetation cover does not, however. The pattern show in the bottom panel of Figure 4 looks like there were a couple outlier plots in year two. My interpretation is that mortality didn’t impact cover in a systematic way, and that there is inherent variation in the system that is driven by some other feature.

Response: We characterized vegetation species richness and cover as a potential covariate of arthropod community structure and found significant variation. It’s unclear to us how the temporal aspects of vegetation cover are interpreted as a reason to dismiss our larger study of arthropods which responded to both vegetation and soil properties—factors that vary in a continuous fashion across the study plots.

Second round: I am not suggesting that the entire study be dismissed because of the vegetation patterns. I am evaluating whether inferences made in the paper are appropriate given the results shown. In my opinion, the data in Figure 4 does not support the statement that tree mortality caused significant changes in vegetation cover.

##

First round: Line 304: I can’t tell what was included in this Mantel test, so it’s difficult to gauge the associated inferences.

Response: This information was listed in detail on lines 245-247 in the methods section describing the Mantel test: “Independent variables used in both Mantel tests and regression models included: soil moisture, pH, %C, %N, DOC, NH4+, microbial biomass, vegetation species richness, total vegetation cover, forb cover, graminoid cover, and tree and shrub cover.”

Second round: The Mantel test is not well explained. There is a single r value reported with a single p-value. There are quite a few variables being examined, however. What variables were actually significant, were partial Mantel test done? Or were all variables used to generate a single Euclidean distance matrix? If so, how were they normalized? In the first round of review I may have missed that Sorensen is used in the Mantel tests. Why is the abundance information not being used here?

##

First round: For the model selection, it would be useful to provide a table of model parameters and statistics. It would be even better to include those statistics for all models that were within about 2 BIC units so the reader can decide how much stock to put into inferences based on the best model.

Response: Model parameters are listed on lines 245-247 of the methods as noted above. The parameters are also listed in the text of Table 3. Model statistics are shown in Table 3 (i.e., the table lists: F-ratio, P-value, adjusted R2, and BIC scores).

Second round: Thank you, I must have missed that previously. Looks like the table needs to be referenced around line 317; also put a table reference near 321. This does not, however, address my point about providing stats on additional models with small BIC deviations. Also, there is no reason to include BIC if there’s only one model displayed as the absolute value of BIC has no meaning, only the relative values matter.

##

First round: Still not understanding the dissimilarity noted on line 308. It is not mentioned in the follow-on results.

Response: This was a mistake, as sample dissimilarity was not analyzed via linear modeling. The reference to dissimilarity has been deleted.

Second round: Thank you.

##

First round: And, there’s no mention of the collinearity tests that were promised in the Methods section.

Response: We performed collinearity tests, as well as tests that data also met assumptions of normality but did not include the output in the manuscript. The availability of the raw data would allow others to run these tests if desired, and these test statistics are rarely included in the results. However, we can supply them as a supplemental if requested by the editor.

Second round: The outcome of the collinearity tests should be reported.

##

First round: I must say I had a very hard time with the Discussion. The authors have a neat sampling design and some tantalizing results, but nothing very definitive at this point. Yet, inferences and implications throughout the Discussion are written in a way that suggests the authors found definitive and major differences through time and across the compartments, and that these differences have serious implications. As summarized above, I don’t see any major differences and I don’t see any strong evidence of deterministic processes. At minimum, I recommend a very significant reduction in the length and speculation of the Discussion. Much better would be reconsidering the results and associated inferences, which would lead to a very different Discussion section.

Response: We respectfully disagree with this assessment, primarily because, as noted above, differences in arthropod community structure, abundance, and diversity were more extensive than described by the reviewer. It would appear this reviewer’s point of view is that arthropod measures within years of our chronosequence representing time since tree mortality can only be compared to measures in the undisturbed control. While this strict interpretation that early and intermediate communities can only be compared to climax communities might be necessary for specific questions, it precludes investigation of successional trajectories in community assembly. Because this type of question has been the focus of much of our recent work and that of our colleagues, this study adds to our collective understanding of processes affecting community assembly over time and not just in relation to a theoretical stable state.

Second round: See my comments above. I still feel there is little convincing evidence and what evidence there is (e.g. community composition through time) is difficult to interpret because year 4 tracks back to year 0. Please also note that I am not suggesting the study should be dismissed, I am suggesting that the authors need to (1) include abundance data throughout, (2) re-do their aboveground analyses with the same level of replication as for belowground, and (3) tone down their inferences to more appropriately reflect their suggestive/interesting but relatively inconclusive results.

·

Basic reporting

no further comments

Experimental design

no further comments

Validity of the findings

no further comments

Additional comments

The authors have responded to reviewer comments thoroughly by making appropriate changes and adding necessary missing detail to clarify confusing issues.

---

## Round 0.3 · accepted · Accept

The paper did not change much from the second round of reviews but I generally agreed with your responses to reviewers. I don't think that further discussion is going to change the content of the paper; both sides have strong opinions and that is just the way it is. So, let's let it sort out, if it will, by publishing the paper and seeing how the broader community responds. I see no fatal flaws in the paper and it does advance understanding of responses of forest arthropods to disturbance.